# HIV Stigma in Awi Zone, Northwest Ethiopia, and a Unique Community Association as a Potential Partner

**DOI:** 10.3390/ijerph21080982

**Published:** 2024-07-27

**Authors:** Muluken Azage Yenesew, Gizachew Yismaw, Dabere Nigatu, Yibeltal Alemu, Addisu Gasheneit, Taye Zeru, Belay Bezabih, Getahun Abate

**Affiliations:** 1School of Public Health, Bahir Dar University, Bahir Dar P.O. Box 79, Ethiopia; muluken.azage@bdu.edu.et (M.A.Y.); dabere.nigatu@bdu.edu.et (D.N.); ybekele@ltu.edu.au (Y.A.); 2Amhara Public Health Institute, Bahir Dar P.O. Box 3898, Ethiopia; gizachewyismaw@aphi.gov.et (G.Y.); taye.z@aphi.gov.et (T.Z.); belay@aphi.gov.et (B.B.); 3Awi Zone Health Office, Amhara Regional State Health Bureau, Bahir Dar P.O. Box 79, Ethiopia; addisu.g@aphi.gov.et; 4Division of Infectious Diseases, Saint Louis University, Saint Louis, MO 63104, USA

**Keywords:** HIV, stigma, Awi, equestrian association

## Abstract

Indigenous institutions play a vital role in fighting HIV stigma by leveraging their cultural knowledge, leadership, and community connections. Understanding HIV/AIDS attitudes, information gaps, and stigma among members of indigenous institutions is critical for devising culturally relevant and successful interventions and preventative strategies. This study was conducted with the objective of assessing the levels of knowledge about HIV/AIDS and the various HIV/AIDS discriminatory attitudes and practices among members of the Awi Equestrian Association, an indigenous association in Awi Zone, Northwest Ethiopia, that plays major roles in the social, cultural, political, and economic activities of the community. The study is a cross-sectional study conducted from June through July 2022. Eight hundred and forty-six people in the study area were interviewed using a pilot-tested questionnaire. Multiple linear regression analysis was used to identify factors associated with the score level of HIV-related stigma. Forty-five percent of study participants did not have adequate knowledge of HIV/AIDS, and 67.4% had moderate to high discriminatory attitudes towards people living with HIV. HIV-stigmatizing practices were high, with 36% admitting to speaking badly about people living with HIV and 23% wanting their relative with HIV to seek treatment in another zone. In our study, low level of knowledge about HIV/AIDS (*p* < 0.001), older age (*p* < 0.05), and male sex (*p* < 0.05) were factors associated with higher levels of stigmatizing practices. In conclusion, HIV-related stigma is common in Awi Zone. The Awi Equestrian Association has become a unique potential partner for HIV control in the area in an effort to achieve United Nation AIDS target of 95–95–95.

## 1. Introduction

Human immunodeficiency virus/Acquired immunodeficiency syndrome (HIV/AIDS) remains a major public health threat worldwide [1,2]. Globally, 38.4 million people were living with HIV in 2021, with 1.5 million patients being newly diagnosed [2]. In the same year, Ethiopia had 612,925 people living with HIV/AIDS (PLWHA) and 11,967 newly diagnosed patients [3]. HIV/AIDS control is challenging but important. The United Nations Program on HIV/AIDS (UNAIDS) has launched the 95–95–95 target with the aims of diagnosing 95% of the PLWHA, initiating treatment on 95% of HIV-diagnosed patients, and achieving viral suppression in 95% of those who are undergoing HIV treatment by 2025 [1,4]. Although progress has been made, the world is far from reaching the 95–95–95 target [2]. 

One of the challenges for HIV/AIDS control is stigma [1,5,6], which is a major problem worldwide [7]. The Center for Disease Control and Prevention defines HIV stigma as negative attitudes and beliefs about people with HIV [8]. It is the prejudice that comes with labeling an individual as part of a group that is believed to be socially unacceptable. Using public domain data in 2016, it was shown that 95% of sexually active Ethiopians had HIV-stigmatizing attitudes [9]. Discriminatory attitudes as high as 80% have been obtained in other sub-Saharan countries [10,11]. HIV-related stigma and discrimination affect the emotional well-being and mental health of PLWHA [8]. Surveys of PLWHA in the north and southern regions of Ethiopia indicated that about 40% perceived stigma as a problem. [12,13]. HIV stigma will lead to self-isolation, community avoidance, and low utilization of HIV and non-HIV specialized healthcare services [6,14,15], leading to a decrease in adherence to anti-retroviral therapy (ART) [1,5]. 

Stigma reduction is challenging because strategies implemented should involve the community, including PLWHA, and have to make meaningful changes in the quality of life of HIV-patients [16]. We chose to focus on Awi Zone of the Amhara region, Ethiopia, because of the presence of a unique indigenous equestrian association which has major influences on the life of the Awi people. The total population of Ethiopia is estimated to be 103 million, and 22.5 million live in the Amhara region [6,17]. It is estimated that there are 612,925 PLWHA in Ethiopia, and 171,555 live in Amhara region [3,6]. Awi Zone is one of the 13 zones of the Amhara region in Northwest Ethiopia, with a total population of about 1 million [18], and has a large influx of internally displaced people (IDP) whose number fluctuates based on political conflicts in the surrounding areas [19]. Sixty percent of people in Awi Zone are of Awi ethnicity and about 40% are of Amhara ethnicity, with 53% speaking Amharic and 45% speaking Awigni as their first language [20]. Of the total new infections of HIV in the Amhara region, the proportion in the year 2020/21 in Awi Zone was 11.3% [21].

It is believed that the Awi Equestrian Association (AEA) was established in the 1940s [22]. AEA serves all people in Awi Zone, and its members include people with Awi and Amhara ethnicity. Most members of the AEA have horses, but owning a horse is not an absolute requirement. The association is involved in organizing social events, including funeral services, marriages, charity for the sick and poor, conflict resolution, land management, efficient and equitable management of cattle, efficient utilization of water resources for irrigation, and leading large annual festivals [22]. In fact, the AEA’s festivals are public events that have tourism potential for the country, and Ethiopia is making an effort to register the AEA’s public festivals as United Nations Educational, Scientific and Cultural Organization (UNESCO) world cultural heritage activities [20]. Because AEA members are household heads, the members are predominantly male, but female participation is always encouraged by the association. This study was conducted with the objective of assessing HIV-related knowledge, attitudes, and practices among members of the AEA.

## 2. Materials and Methods

### 2.1. Study Settings

The study was conducted in Awi Zone, the Amhara region, Northwest Ethiopia. The zone borders the Benishangul-Gumuz region on the west, North Gondar Zone on the north, and West Gojjam on the east. The administrative town of the zone is Injibara, which is located 440 km from Addis Ababa, the capital city of Ethiopia, and 120 km from Bahir Dar, the capital city of the Amhara region. The zone has nine districts and seven town administrations, with a total population of 1,159,386, of which only 12.5% are urban inhabitants [23]. 

### 2.2. Study Design and Population

A cross-sectional study design was conducted from June to July 2022 to assess the knowledge, attitude, and practice of HIV/AIDS-related stigma among members of the AEA living in Awi Zone. The AEA leaders were consulted about the survey methodology. During the study period, the association had more than 65,000 members living in different districts of the zone. Members of the association were the study population. 

### 2.3. Sample Size and Sampling Techniques

A single population formula [n = Z_α/2_^2^ × *p* × (1 − *p*)/d^2^] was used to determine the sample size by considering the following assumptions: 50% of proportions having appropriate knowledge, positive attitudes, and practices since there was no previous study (*p* = 50%), 95% confidence level (Z_α/2_ = 1.96), and 5% of margin of error (d = 0.05). The calculated sample size was 853, considering 2 design effects and a 10% non-response rate. Four districts and four town administrations were randomly selected from all districts and town administrations in the zone. Proportion to size allocation was made to determine the required sample size from selected districts and towns. A simple random sampling technique was used to select members of the association using the list of each selected district and town as a sampling frame. It was expected that female participation may be low because only a few families were known to have a woman as the head of the household, and, therefore, an effort was made to increase female participation by trying to enroll all the female household heads in the area. 

### 2.4. Data Collection Tool

A structured questionnaire was used to collect data. The questionnaire was developed by reviewing similar literature [3,24,25,26,27,28,29,30] and comprised the following five sections: (i) socio-demographic characteristics, (ii) knowledge related to HIV/AIDS, (iii) attitude towards HIV/AIDS, (iv) HIV/AIDS-related stigma, and (v) discriminatory practice and HIV/AIDS-related stigma prevention strategies. The knowledge part of the questionnaire had 18 items, the attitude had 13 items, HIV-related stigma had 13 items, and discriminatory practice and HIV/AIDS-related stigma prevention strategies had 12 items. 

The questionnaire was first prepared in English and then translated to Amharic by two co-authors who know the customs of the Awi people (M.A. and G.Y.). The questionnaire was reviewed by all co-authors for consistency and correctness. The questionnaire was pre-tested and necessary modifications were made to make the question clear. Data collectors and supervisors were trained on the objective of the study and on how to interview study participants. The questionnaire was completed on paper. The survey was conducted anonymously. All participants were offered a chance to complete the questionnaire themselves. For participants who could not read and write, all questions in the questionnaire were read in Amharic and their responses were documented in the questionnaire by the study team. Their responses to each question were read to them for verification of accuracy. Data were entered into a web-based ODK site and checked for completeness by supervisors daily.

Bloom’s cut-off scoring system was used to score knowledge about HIV/AIDS [31]. Briefly, the correct responses were scored as 1 while the incorrect responses were scored as 0. The maximum obtainable knowledge score was 100% (i.e., 18 correct answers); scores ≤ 50% (i.e., ≤9 correct answers) were classified as low knowledge, 51–74% (i.e., 10–13 correct answers) were moderate, and scores ≥ 75% (i.e., at least 14 correct answers) were classified as high knowledge. Attitude and practice were scored as described previously [28,32,33]. Briefly, the correct responses on attitude were scored as 1 while the incorrect responses were scored as 0. The maximum obtainable ‘attitude’ score was 12; scores 0–5 were classified as high discriminatory attitudes towards HIV/AIDS patients, 6–8 as moderate, 9–11 as low discriminatory attitude, and 12 as nondiscriminatory attitude. Similarly, the correct responses on ‘practice’ were scored as 1 while the incorrect responses were scored as 0. The maximum obtainable ‘practice’ score was 9. A score of 0–3 was classified as high discriminatory practice, 4–8 as low discriminatory practice, and 9 as non-discriminatory practice. 

### 2.5. Data Management and Analysis

Data were extracted from the web-based data collection tool and exported to SPSS version 20 for analysis. Descriptive statistics were used to describe socio-demographic characteristics and other variables. Wealth status was scored using principal component analysis. Wealth status was assessed as an indicator of socioeconomic status and was computed by principal component analysis from variables used in an Ethiopian Demographic Health survey based on owning farmland, having a house with a toilet, owning a house with corrugated iron roofing, a bank account, having a mobile phone, electricity, a number of cows and oxen, a number of horses/mules/donkeys, and a number of goats/sheep and chicken [34]. Participants were divided into tertiles (i.e., poor, medium and rich). 

Binary and multivariable logistic regression models were used to analyze the association between independent variables and outcome variables. Chi-square assumptions and the Hosmer–Lemeshow test were checked to assess the model fitted to conduct logistic regression [35]. The adjusted odds ratio (AOR), with the corresponding 95 % confidence interval (CI), was calculated to identify factors associated with outcome variables. A *p*-value < 0.05 was considered statistically significant. 

### 2.6. Ethical Considerations

Ethical approval was obtained from the Ethical Review Board of the Amhara Public Health Institute. A support letter was given for zonal and district AEA office heads. Consent was obtained from the study participants after explaining the objectives of the study. Study participants were informed about their right to withdraw from the interview at any time. All records were kept confidential, and confidentiality was further assured by avoiding the collection of personal identifiers.

## 3. Results

### 3.1. Socio-Demographic Characteristics

A total of 846 participants were included in the study, with a 99.2% response rate. Two-hundred and fifty-five (30.1%) participants were AEA leaders, and the remaining were AEA members. One-third of the participants were 40 years to 49 years old and 97% were males. Nearly 30% of the participants lived in urban areas and 95% were married. Two hundred and thirty-six (28%) and 446 (53%) participants and their partners could not read and write, respectively. Six hundred and one (71%) participants and 580 (69%) of their partners were farmers (Table 1). One-third of participants were considered rich and one-third were poor.

### 3.2. Knowledge of HIV/AIDS

Table 2 shows the results of knowledge about HIV/AIDS. Of the 18 questions used to assess knowledge about HIV/AIDS, five received incorrect responses from more than 20% of respondents. These were not knowing HIV has treatment (46.8%), not knowing HIV can be transmitted to newborns via breastfeeding (37.8%), believing that sex with a virgin cures HIV (35.2%), and not knowing about HIV transmission to the fetus during pregnancy (34.4%) and delivery (25.3%). An overall score of ≤50% (i.e., ≤9 of correct answers) defines low knowledge, 51–74%, moderate, and ≥75% high knowledge levels. More than half of the study participants (54.7% with 95%CI 51.4% to 58.1%) showed a high knowledge level, while 35.8% (95%CI 32.6% to 39.1%) had moderate and 9.5% (95%CI 7.7% to 11.6%) had low knowledge levels.

### 3.3. Attitude of Study Participants toward HIV/AIDS

The most common stigmatizing attitude was believing that a person with HIV/AIDS must have done something wrong and deserves to be punished (47.2%) followed by the notion that HIV status should be revealed to family members regardless of patient’s consent (43.3), HIV-positive women should not get pregnant (38.8%), not feeling comfortable sharing a plate with HIV-positive person (34.8%), and believing that PLWHA must expect restrictions on their freedom (22.3%). One-third of study participants (32.6%) had a low discriminatory attitude, 38.4% had a moderate discriminatory attitude, and 29% had a high discriminatory attitude towards HIV/AIDS patients (Table 3).

### 3.4. HIV/AIDS-Related Practice

The practices of study participants are shown in Table 4. Five hundred and eleven (60.4%) believed that people hesitate to take an HIV test because they are afraid of how other people would react if the test was positive, and 30.3% believed that PLWHA lose the respect of other people. The most common stigmatizing practice was talking badly about PLWHA (35.9%), followed by being ashamed if someone in the family is diagnosed with HIV (30.3%), changing interactions if a family member is diagnosed with HIV (26.5%), and not showing willingness to work in a committee with a HIV-diagnosed person (25.9%). Overall, 74.9% (95%CI 71.9–77.8%) of participants had a low or high level of HIV/AIDS stigmatizing practices.

### 3.5. Factors Associated with HIV/AIDS-Knowledge, Attitude and Related Stigma

Young age, being married, and better educational status were significantly associated with higher knowledge about HIV/AIDS in the binary logistic regression analysis. A multivariable logistic regression model showed that the ability to read and write, formal education, and being married increased the odds of having higher knowledge about HIV/AIDS. Young age was significantly associated with higher knowledge scores (Table 5). 

### 3.6. Factors Associated with Discriminatory Attitude

In the binary logistic regression analysis, knowledge about HIV/AIDS, older age, low educational status and living in a rural area, and poor wealth status increased the odds of having a discriminatory attitude (Table 6). Low knowledge level about HIV/AIDS, older age, and male sex were significantly associated with discriminatory attitudes in a multivariable logistic analysis. Participants with lower and moderate knowledge about HIV/AIDS had higher odds of having a discriminatory attitude compared to those with a higher knowledge. Older age and male sex had higher odds of having a discriminatory attitude compared to younger age and female sex, respectively (Table 6).

### 3.7. Factors Associated with HIV-Stigmatizing Practices

The binary logistic regression analysis showed that attitudes against PLWHA, knowledge about HIV/AIDS, older age, being a farmer, and being poor increased the odds of having HIV-stigmatizing practices (Table 7). A multivariable logistic regression model showed that moderate and higher discriminatory attitudes are associated with higher odds of having HIV-related stigmatizing practices compared to lower discriminatory attitudes. Male sex was significantly more associated with HIV-stigmatizing practices compared to female sex (Table 7). 

## 4. Discussion

This study on members of the AEA shows that 45.3% of participants lack adequate knowledge about HIV/AIDS, 67% had moderate or high discriminatory attitudes, and 74.9% reported practices that stigmatized HIV/AIDS patients. These findings in members of an association with major social influence and in a region where HIV incidence remains high are alarming, making stigma reduction, with a focus on the AEA, crucial in the fight against HIV/AIDS in Awi Zone, Ethiopia. Our results showed that 97% of participants were male. This is not surprising, because only 5.5% of the AEA members are female.

Among the responses to the 18 questions used to assess HIV/AIDS knowledge levels, the response about ‘having sex with a virgin’ as a way of curing HIV surprisingly had affirmative responses from one-third of participants. The misconception that having sex with a virgin cures HIV has been reported in previous studies in Africa [36,37]. In South Africa, this misconception has been identified as a possible factor in the rape of babies and children [36]. This sexual practice between older HIV-infected patients and virgins who are very young and vulnerable increases the risk of HIV transmission [38]. The perception is even more concerning in Awi Zone, which gives shelter to many IDP [39]. Adolescent girls who are displaced or refugees, particularly those who are more accepting of gender inequitable norms, have higher HIV risk factors [40]. A study on IDP in the Democratic Republic of Congo found that HIV prevalence is higher among women who are IDP compared to non-IDP women in the same area (7.6% vs. 3.1%) [41]. 

Nearly half of study participants did not have adequate knowledge about HIV/AIDS, higher than the results obtained from a population-based study in Ethiopia which showed inadequate knowledge in only 29% of participants [26]. The reason for this difference is not clear. Our study showed that knowledge about HIV/AIDS is higher in participants with formal education, an ability to read and write, and with a married status. These findings are similar to the findings obtained in other studies from Africa, Asia, and South America [29,30,42]. Inadequate knowledge about HIV/AIDS has been shown to have a strong association with lower educational level and higher rates of HIV stigma [43,44,45]. In this study, older age was significantly associated with lower HIV/AIDS knowledge scores, unlike findings from previous studies [24,30,46]. The difference could be attributed to the opportunity for more exposure to health education, which older patients in our study may not have had [47].

Our study revealed that 67.4% of participants had a moderate or high discriminatory attitude towards PLWHA. This finding of high levels of discriminatory attitudes is consistent with the overall findings from the national demographic and health survey in Ethiopia [26]. Similar levels of discriminatory attitudes were also obtained from studies in Nigeria and South Africa [48,49]. High levels of discriminatory attitudes towards PLWHA have dire consequences, including emotional stress, inconsistent health-care-seeking behavior, non-disclosure of HIV status, inadequate self-care, late initiation of ART, poor adherence to ART, and suboptimal utilization of social support [25,50,51,52]. Our findings indicate that low levels of knowledge about HIV/AIDS, older age, and male sex were associated with higher discriminatory attitudes. Low level of knowledge about HIV/AIDS and male sex were previously shown to be associated with higher discriminatory attitudes in a study involving 15 sub-Saharan African countries [29]. In the same study, contrary to our findings, younger age was associated with higher discriminatory practices [29]. Although the reason for this difference is not clear, internet availability and access to online educational materials is expected to make younger age groups have lower discriminatory attitudes. Data from the UNICEF multiple indicator cluster surveys collected from Ghana, Guinea Bissau, Malawi, and Zimbabwe showed that participants who reported ever using a computer and the internet were more likely to have higher HIV/AIDS knowledge compared to those who did not [53].

We show that 75% of participants in our study admitted to HIV stigmatizing practices. This is concerning, as these stigmatizing practices may increase resistance in regard to HIV testing and decrease the willingness of PLWHA, including newly diagnosed patients, to utilize available resources [43]. Studies from other parts of Ethiopia and Asia have shown that higher discriminatory attitudes increased the odds of having HIV-related stigmatizing practices similar to the findings in our study [24,27]. In our study, in addition to moderate and higher discriminatory attitudes, male sex was associated with higher odds of HIV-stigmatizing practices. This may be reflective of a male-dominant culture in the study area.

Lack of adequate knowledge, discriminatory attitudes, and stigmatizing practices can potentially contribute to HIV spread and poor outcomes, affecting the HIV control efforts aimed at the 95–95–95 target [54,55].This study is unique in that it assessed HIV stigma among members of a community association that has a major influence on the life of the Awi people. 

This study may have some limitations: (1) The majority of participants were male and, therefore, the results may not reflect HIV stigma among females in the population. In future works on HIV stigma in the area, we will work with AEA leaders to find mechanisms to effectively recruit female participants in ways that reflect gender distribution in the area. (2) A lack of a more clear or universal definition of discriminatory attitudes and HIV-related stigma, which might underestimate the magnitude of stigma and discriminatory attitudes in the community. (3) The quality of the data may have been affected by social desirability bias, since study participants may not express real attitudes and HIV-related stigma during interviews. (4) The investigation of the directionality of the relationships between HIV stigma and associated factors was not possible due to the cross-sectional design.

## 5. Conclusions

The study revealed that inadequate knowledge, discriminatory attitudes, and HIV-stigmatizing practices were common among AEA members. The AEA could be a target for HIV/AIDS health education and a potential partner for HIV stigma reduction and HIV control in the area. Collaborating with the AEA is essential for developing effective and sustainable strategies to address HIV stigma and improve the lives of individuals and communities affected by HIV/AIDS. Future studies will include the health education of AEA members and impact assessment using changes in the number of newly HIV-diagnosed patients, the proportion of HIV-diagnosed patients on treatment, and the proportion of patients with HIV viral load suppression as endpoints.

## Figures and Tables

**Table 1 ijerph-21-00982-t001:** Socio-demographic characteristics of survey respondents among members of AEA in 2022.

Variable		Frequency	Percentage (%)
Age	20–39	204	24.1
	40–59	494	58.4
	≥60	148	17.5
Sex	Male	818	96.7
	Female	28	3.3
Place of residence	Urban	251	29.7
	Rural	595	70.3
Role in the association	Leader	255	30.1
	Member	591	69.9
Marital status	Married	803	94.9
	Single/divorced/separated/widowed	43	5.1
Educational status	Cannot read and write	236	27.9
	Can read and write	242	28.6
	Primary education	232	27.4
	Secondary education and above	136	16.1
Educational status of partner (*n* = 803)	Cannot read and write	446	55.5
	Can read and write	178	22.2
	Primary education	123	15.3
	Secondary education and above	46	7.0
Occupation	Farmer	601	71.0
	Merchant	114	13.5
	Employee	59	7.0
	Daily laborer	42	5.0
	Un-employee	12	1.4
	Other ^B^	18	2.1
Occupation of partner	Farmer	580	68.6
	Employee	30	3.6
	Daily laborer	37	4.4
	Un-employee	90	10.6
	Other ^B^	66	7.8
Wealth status	Poor	282	33.3
	Medium	286	33.8
	Rich	278	32.9

^B^ waver, plumber and local beer maker.

**Table 2 ijerph-21-00982-t002:** Knowledge of HIV/AIDS of survey respondents among members of AEA in 2022.

Variables	Yes	%
Ever heard of HIV or AIDS	840	99.3
A person can get HIV/AIDS by sexual intercourse with someone who has HIV/AIDS	818	96.7
A person can get HIV/AIDS by living together with someone who has HIV/AIDS	159	18.8
People can get HIV by sharing food with a person who has HIV	133	15.7
People can get HIV because of witchcraft or other supernatural means	125	14.8
It is possible for a healthy-looking person to have HIV	705	83.3
You can get HIV/AIDS by touching someone who has HIV/AIDS	165	19.5
Men can transmit HIV/AIDS to women	834	98.6
Women can transmit HIV/AIDS to men	835	98.7
A pregnant woman can contract HIV/AIDS to her baby during pregnancy	555	65.6
A pregnant woman can contract HIV/AIDS to her baby during childbirth/delivery	632	74.7
A woman can contract HIV/AIDS to her baby by breastfeeding	526	62.2
A person can get rid of HIV/AIDS by having sex with a virgin	298	35.2
There is a cure for HIV/AIDS	66	7.8
Knowing a person cured of HIV with traditional medicine	38	4.5
HIV has treatment	450	53.2
Knowing a person cured of HIV with holy water	112	13.2
Patients with HIV can live a normal life	699	82.6
Overall knowledge levels		
Higher knowledge	463	54.7
Moderate knowledge	303	35.8
Lower knowledge	80	9.5

**Table 3 ijerph-21-00982-t003:** Attitudes of study participants toward HIV/AIDS patients in 2022.

Variable	Agree	DNK	Disagree
Believe that people with HIV/AIDS are cursed	89 (10.5)	6 (0.7)	751 (88.8)
Believe that people with HIV/AIDS are dirty	77 (9.1)	2 (0.2)	767 (90.7)
Believe that people with HIV/AIDS should be ashamed	173 (20.5)	11 (1.3)	662 (78.3)
Believe that people with HIV/AIDS must expect some restrictions on their freedom	189 (22.3)	38 (4.5)	619 (73.2)
Believe that a person with HIV/AIDS must have done something wrong and deserves to be punished	399 (47.2)	36 (4.3)	411 (48.6)
Believe that people with HIV should be isolated	105 (12.4)	19 (2.3)	722 (85.3)
Not wanting to be a friend with someone who has HIV/AIDS	188 (22.2)	13 (1.5)	645 (76.3)
Believe that people with HIV/AIDS should not be allowed to work	150 (17.7)	8 (1.0)	688 (81.3)
When a patient is tested positive, the HIV status should be disclosed to very close relatives regardless of patient’s consent	366 (43.3)	48 (5.7)	432 (51.0)
Believe that people with HIV/AIDS should not get married as HIV/AIDS has no cure	245 (29.0)	18 (2.1)	583 (68.9)
HIV-positive women should not get pregnant as they know that HIV/AIDS has no cure and they will eventually die	328 (38.8)	24 (2.8)	494 (58.4)
Feeling comfortable if you share a plate with a HIV-diagnosed person	532 (62.9)	19 (2.3)	295 (34.8)
Mean score (%)	236 (27.9)	20 (2.4)	588 (69.6)

**Table 4 ijerph-21-00982-t004:** HIV-related practice of survey respondents among members of AEA in 2022.

Variables	Yes (%)	No (%)
You allow a person with HIV to work with you	736 (87.0)	110 (13.0)
You decrease your interaction if a friend is diagnosed with HIV	264 (31.2)	582 (68.8)
You change your interaction if your sister, brother or parent is diagnosed with HIV	224 (26.5)	622 (73.5)
You object for HIV-positive person to serve on a committee with you	219 (25.9)	627 (74.1)
You buy fresh vegetables from a shopkeeper or vendor if you knew that this person had HIV	650 (76.8)	196 (23.8)
Children living with HIV should be allowed to attend school with children who do not have HIV	721 (85.2)	125 (14.8)
People hesitate to take an HIV test because they are afraid of how other people will react if the test result is positive for HIV	511 (60.4)	335 (39.6)
You talk badly about people living with HIV, or who are thought to be living with HIV	304 (35.9)	542 (64.1)
People living with HIV, or thought to be living with HIV, lose the respect of other people	256 (30.3)	590 (69.7)
Are you ashamed if someone in your family had HIV	295 (34.7)	551 (65.1)
HIV patients have the same burial site as other people	770 (91.0)	76 (9.0)
You will feel better if your family with HIV/AIDS can visit health facilities within the zone	656 (77.5)	190 (22.5)
A family member with HIV/AIDS should not visit a health facility within the zone	683 (80.7)	163 (19.3)
Mean score (%)	483.7 (57.2)	362 (42.8)

**Table 5 ijerph-21-00982-t005:** Factors associated with HIV/AIDS knowledge among members of the Awi Equestrian Association in 2022.

Variable		COR (95%CI)	AOR (95%CI)
Age	20–39	2.36 (1.53–3.63) ***	1.73 (1.09–2.76) *
	40–59	1.56 (1.07–2.25) *	1.44 (0.97–2.13)
	≥60	1.00	1.00
Sex	Male	1.91 (0.88–4.13)	0.91 (0.34–2.47)
	Female	1.00	1.00
Place of residence	Urban	1.09 (0.81–1.46)	0.91 (0.56–1.47)
	Rural	1.00	1.00
Role in the association	Leader	1.27 (0.94–1.71)	1.02 (0.74–1.40)
	Member	1.00	1.00
Marital status	Married	5.72 (2.62–12.49) ***	4.87 (1.94–12.22) **
	Single/divorced/separated/widowed	1.00	1.00
Educational status	Cannot read and write	1.00	1.00
	Can read and write	2.23 (1.55–3.22) ***	2.25 (1.52–3.31) ***
	Primary education	2.80 (1.93–4.08) **	2.45 (1.64–3.66) ***
	Secondary education and above	3.52 (2.25–5.49)	3.43 (2.04–5.77) ***
Occupation	Farmer	1.00	1.00
	Merchant	1.46 (0.96–2.20)	1.18 (0.64–2.17)
	Employee	1.01 (0.59–1.72)	0.70 (0.34–1.44)
	Daily laborer	0.64 (0.34–1.20)	0.60 (0.29–1.26)
	Other ^B^	0.97 (0.47–2.02)	0.88 (0.36–2.16)
Wealth status	Poor	1.00	1.00
	Medium	1.22 (0.87–1.69)	0.96 (0.66–1.40)
	Rich	1.28 (0.92–1.79)	1.00 (0.70–1.43)

^B^ waver, plumber and local beer maker. * *p* < 0.05, ** *p* < 0.01, *** *p* < 0.001.

**Table 6 ijerph-21-00982-t006:** Binary and multivariable logistic regression on factors associated with discriminatory attitude towards HIV/AIDS.

Variable		COR (95%CI)	AOR (95%CI)
Knowledge about HIV/AIDS	Low knowledge	12.35 (7.22–21.13) ***	10.51 (5.88–18.78) ***
	Moderate knowledge	5.58 (3.97–7.83)	5.35 (3.75–7.61) **
	High knowledge	100	1.00
Age	20–39	0.38 (0.24–0.60) ***	0.57 (0.33–0.97) *
	40–59	0.53 (0.36–0.77) ***	0.68 (0.44–1.04)
	≥60	1.00	1.00
Sex	Male	1.81 (0.72–4.51)	3.62 (1.05–12.50) *
	Female	1.00	1.00
Place of residence	Urban	0.64 (0.46–0.88) **	0.75 (0.42–1.34)
	Rural	1.00	1.00
Role in the association	Leader	0.92 (0.67–1.26)	1.17 (0.81–1.69)
	Member	1.00	1.00
Marital status	Married	0.40 (0.22–0.74) **	0.46 (0.19–1.15)
	Single/divorced/separated/widowed	1.00	1.00
Educational status	Cannot read and write	2.52 (1.58–4.00) ***	1.12 (0.61–2.03)
	Can read and write	1.04 (0.64–1.68)	0.77 (0.43–1.39)
	Primary education	1.17 (0.73–1.89)	1.01 (0.57–1.79)
	Secondary education and above	1.00	1.00
Occupation	Farmer	1.00	1.00
	Merchant	0.37 (0.22–0.64) ***	0.60 (0.28–1.31)
	Employee	0.82 (0.46–1.46)	1.30 (0.56–3.00)
	Daily laborer	1.54 (0.82–2.89)	1.78 (0.79–4.07)
	Other ^B^	0.93 (0.43–2.03)	1.07 (0.37–3.03)
Wealth status	Poor	1.00	1.00
	Medium	0.49 (0.34–0.70) ***	0.61 (0.39–0.94) *
	Rich	0.65 (0.46–0.92) *	0.73 (0.48–1.09)

^B^ waver, plumber and local beer maker. * *p* < 0.05, ** *p* < 0.01, *** *p* < 0.001.

**Table 7 ijerph-21-00982-t007:** Binary and multivariable logistic regression on factors associated with HIV-related stigmatizing practice.

Variable		COR (95%CI)	AOR (95%CI)
Attitude towards to HIV	Low discriminatory attitude	1.00	1.00
	Moderate discriminatory attitude	2.71 (1.90–3.87) ***	2.73 (1.85–4.02) ***
	high discriminatory attitude	13.48 (7.77–23.39) ***	12.64 (6.79–23.51) ***
Knowledge on HIV/ADIS	Low knowledge	5.29 (2.25–12.45) ***	1.94 (0.73–5.14)
	Moderate knowledge	1.51 (1.08–2.12) *	0.80 (0.53–1.21)
	High knowledge	1.00	1.00
Age	20–39	1.00	1.00
	40–59	0.79 (0.54–1.15)	0.72 (0.46–1.13)
	≥60	1.81 (1.04–3.15) *	1.30 (0.69–2.45)
Sex	Male	5.79 (2.63–12.75) ***	5.22 (1.66–16.46) **
	Female	1.00	1.00
Place of residence	Urban	0.45 (0.32–0.62)	0.53 (0.30–0.94) *
	Rural	1.00	1.00
Role in the association	Leader	0.65 (0.47–0.90)	0.69 (0.47–1.01)
	Member	1.00	1.00
Marital status	Married	1.48 (0.76–2.85)	1.18 (0.39–3.56)
	Single/divorced/separated/widowed	1.00	1.00
Educational status	Cannot read and write	1.60 (0.96–2.66)	0.84 (0.43–1.63)
	Can read and write	0.70 (0.44–1.12)	0.74 (0.41–1.33)
	Primary education	1.12 (0.68–1.82)	1.09 (0.60–1.96)
	Secondary education and above	1.00	1.00
Occupation	Farmer	1.00	1.00
	Merchant	0.32 (0.21–0.48) ***	0.67 (0.34–1.34)
	Employee	1.08 (0.56–2.10)	1.84 (0.72–4.69)
	Daily laborer	0.78 (0.38–1.59)	0.73 (0.30–1.79)
	Other ^B^	1.10 (0.44–2.76)	1.67 (0.54–5.23)
Wealth status	Poor	1.00	1.00
	Medium	0.61 (0.42–0.89) *	0.95 (0.60–1.52)
	Rich	1.02 (0.69–1.53)	1.12 (0.70–1.74)

^B^ waver, plumber and local beer maker. * *p* < 0.05, ** *p* < 0.01, *** *p* < 0.001.

## Data Availability

The research data are available with the first and last authors and will be available upon request.

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
