# Peer review of "HIV Stigma in Awi Zone, Northwest Ethiopia, and a Unique Community Association as a Potential Partner"

_ijerph, 2024, doi:10.3390/ijerph21080982_

Round 1

Reviewer 1 Report (Previous Reviewer 1)

Comments and Suggestions for Authors

1. You should add the overall mean of attitude and practice for both table 3 and 4.

2. Lines 183-185: delete the sentence :" Overall HIV/AIDS knowledge....correct answers."

3. Lines 199-202: delete :" to calculate....low descriminatory attitude ".

4.Lines 213-216: delete: "overall...respectively".

5. Line 43: correct as: On of the challenges....(1,5,6) which is.....  (7).

Comments on the Quality of English Language

Minor editing required

Author Response

Point-by-point reply to reviewers’ comments

Reviewer 1

1. You should add the overall mean of attitude and practice for both table 3 and 4.

            Overall mean added to both tables.

2. Lines 183-185: delete the sentence :" Overall HIV/AIDS knowledge....correct answers."

            Sentence deleted as suggested.

3. Lines 199-202: delete :" to calculate....low descriminatory attitude ".

            Sentence deleted as suggested.

4. Lines 213-216: delete: "overall...respectively".

            Sentence deleted as suggested.

5. Line 43: correct as: On of the challenges....(1,5,6)whichis.....  (7).

            Corrected as suggested.

Reviewer 2 Report (New Reviewer)

Comments and Suggestions for Authors
  1. Unless the study population is representative of Northwest Ethiopia (which the authors included in the study title), I suggest the authors rephrase the study title as appropriate. This is especially given that males constitute 96.7% of the current study participants, which we can all agree is not the picture for the Northwest Ethiopia region.

  2. Furthermore, can this proportion of male participants in the study also count as one limitation? 

  3. In your analysis sub-section where you mentioned the Hosmer–Lemeshow, please cite the reference for this test.

  4. Some areas in this manuscript appear dull relative to the rest of the text, indicating a copy from another document. Please review this to ensure that uniform color.

  5.  

Best

Bes

Author Response

Point-by-point reply to reviewers’ comments

Reviewer 2

1. Unless the study population is representative of Northwest Ethiopia (which the authors included in the study title), I suggest the authors rephrase the study title as appropriate. This is especially given that males constitute 96.7% of the current study participants, which we can all agree is not the picture for the Northwest Ethiopia region.

            The title is modified based on comment.

2. Furthermore, can this proportion of male participants in the study also count as one limitation?

            This is included as one of the limitations of the study.

3. In your analysis sub-section where you mentioned the Hosmer–Lemeshow, please cite the reference for this test.

            A new reference is added.

4. Some areas in this manuscript appear dull relative to the rest of the text, indicating a copy from another document. Please review this to ensure that uniform color.

            The manuscript now has uniform color.

This manuscript is a resubmission of an earlier submission. The following is a list of the peer review reports and author responses from that submission.

Round 1

Reviewer 1 Report

Comments and Suggestions for Authors

I would thank the authors for their efforts to improve the quality of the manuscript. I have just somme comments that should be improved:

Major comments: 

The descriptions of tables 2, 3, and 4 in the results is still very simplistic ans superficial. You should complete the description of table 2 with the most important findings regarding each item, as some of them was discussed. In addition, the description of tables 1 and 2 has no relation with the content of the tables (how can I find low or moderate discriminatory attitude in table 3 or that they practice stigmatization in table 4?). complete the table and the description as requested for table 2.

The second concern is related to the introduction, I recognize that it has been really improved. however, you should try to delete the repetition in some parts (Example : from line 41 to 56 the term HIV stigma  is repeated multiple times, try to reformulate the sentences to avoid the repetitions).

Minor comments:

Line 16: Complete the objectives (knowledge, descriminatory......)

Line 25, 26...: p value should be in lower case.

Line 41: the last sentence (one...(1,5,6)) should be associated with the second paragraph.

Table 2: add the value of the mean of knowledge

Tables 2, 3, and 4: complete as requested before (see the comment above). 

Try to improve the quality of your tables 

Standardize the use of the numbers in tables 5, 6 and 7 (some case are empty, the reference (1.00) should be standardized in all these tables.

Table 5: in the title, it should be knowledge not stigma

Correct the column related to the wealth status (rich)

Line 245:  what does IDP mean?

Line 304: How can the AEA member be a potential partner since you find that they practice stigmatisation. They should be sentitized before they can be involved.

Comments on the Quality of English Language

Minor editing required

Author Response

Point by point response to reviewer comments

Major comments:

The descriptions of tables 2, 3, and 4 in the results are still very simplistic and superficial. You should complete the description of table 2 with the most important findings regarding each item, as some of them was discussed. In addition, the description of tables 1 and 2 has no relation with the content of the tables (how can I find low or moderate discriminatory attitude in table 3 or that they practice stigmatization in table 4?). complete the table and the description as requested for table 2. The description of tables 1-4 revised.

The second concern is related to the introduction, I recognize that it has been really improved. however, you should try to delete the repetition in some parts (Example : from line 41 to 56 the term HIV stigma is repeated multiple times, try to reformulate the sentences to avoid the repetitions). The introduction section is revised to avoid reptition.

Minor comments:

Line 16: Complete the objectives (knowledge, descriminatory......) Modified as suggested.

Line 25, 26...: p value should be in lower case. Modified as suggested.

Line 41: the last sentence (one...(1,5,6)) should be associated with the second paragraph. Modified as suggested.

Table 2: add the value of the mean of knowledge Overall scores are provided as used by others. A brief explanation how the scores were calculated is included in the result section.

Tables 2, 3, and 4: complete as requested before (see the comment above). Modified as suggested.

Try to improve the quality of your tables Tables revised. We can move tables 2-4 to supplement.

Standardize the use of the numbers in tables 5, 6 and 7 (some case are empty, the reference (1.00) should be standardized in all these tables. Tables revised.

Table 5: in the title, it should be knowledge not stigma Modified as suggested.

Correct the column related to the wealth status (rich) It is not clear what the reviewer comment is referring to but column values checked.

Line 245: what does IDP mean? IDP refers to internally displaced people. The abbreviation and full meaning were first used in the introduction section.

Line 304: How can the AEA member be a potential partner since you find that they practice stigmatisation. They should be sentitized before they can be involved. The conclusion is modified and health education to AEA members included as a need.

Comments on the Quality of English Language

Minor editing required Manuscript reviewed and edited.